# Is ChatGPT Good at Search?
# Investigating Large Language Models as Re-Ranking Agents

**Weiwei Sun**[1*]  **Lingyong Yan**[2]  **Xinyu Ma**[2]  **Shuaiqiang Wang**[2]
**Pengjie Ren**[1]  **Zhumin Chen**[1]  **Dawei Yin**[2†]  **Zhaochun Ren**[3†]
[1]Shandong University, Qingdao, China    [2]Baidu Inc., Beijing, China
[3]Leiden University, Leiden, The Netherlands
{sunnweiwei,lingyongy,xinyuma2016,shqiang.wang}@gmail.com
{renpengjie,chenzhumin}@sdu.edu.cn    yindawei@acm.org
z.ren@liacs.leidenuniv.nl

## Abstract

Large Language Models (LLMs) have demonstrated remarkable zero-shot generalization across various language-related tasks, including search engines. However, existing work utilizes the generative ability of LLMs for Information Retrieval (IR) rather than direct passage ranking. The discrepancy between the pre-training objectives of LLMs and the ranking objective poses another challenge. In this paper, we first investigate generative LLMs such as ChatGPT and GPT-4 for relevance ranking in IR. Surprisingly, our experiments reveal that properly instructed LLMs can deliver competitive, even superior results to state-of-the-art supervised methods on popular IR benchmarks. Furthermore, to address concerns about data contamination of LLMs, we collect a new test set called NovelEval, based on the latest knowledge and aiming to verify the model's ability to rank unknown knowledge. Finally, to improve efficiency in real-world applications, we delve into the potential for distilling the ranking capabilities of ChatGPT into small specialized models using a permutation distillation scheme. Our evaluation results turn out that a distilled 440M model outperforms a 3B supervised model on the BEIR benchmark. The code to reproduce our results is available at www.github.com/sunnweiwei/RankGPT.

## 1 Introduction

Large Language Models (LLMs), such as ChatGPT and GPT-4 (OpenAI, 2022, 2023), are revolutionizing natural language processing with strong zero-shot and few-shot generalization. By pre-training on large-scale text corpora and alignment fine-tuning to follow human instructions, LLMs have demonstrated their superior capabilities in language understanding, generation, interaction, and reasoning (Ouyang et al., 2022).

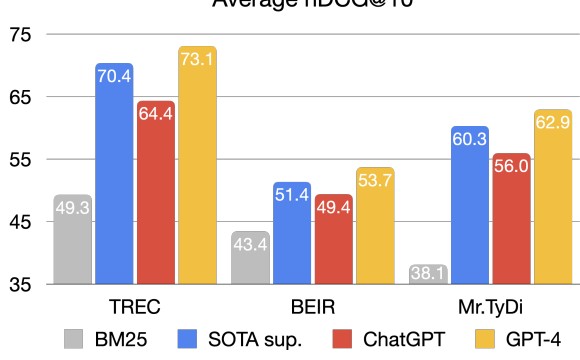

Figure 1: Average results of ChatGPT and GPT-4 (zero-shot) on passage re-ranking benchmarks (TREC, BEIR, and Mr.TyDi), compared with BM25 and previous best-supervised systems (SOTA sup., e.g., monoT5 (Nogueira et al., 2020)).

As one of the most successful AI applications, Information Retrieval (IR) systems satisfy user requirements through several pipelined sub-modules, such as passage retrieval and re-ranking (Lin et al., 2020). Most previous methods heavily rely on manual supervision signals, which require significant human effort and demonstrate weak generalizability (Campos et al., 2016; Izacard et al., 2022). Therefore, there is a growing interest in leveraging the zero-shot language understanding and reasoning capabilities of LLMs in the IR area. However, most existing approaches primarily focus on exploiting LLMs for content generation (e.g., query or passage) rather than relevance ranking for groups of passages (Yu et al., 2023; Microsoft, 2023).

Compared to the common generation settings, the objectives of relevance re-ranking vary significantly from those of LLMs: the re-ranking agents need to comprehend user requirements, globally compare, and rank the passages based on their relevance to queries. Therefore, leveraging the LLMs' capabilities for passage re-ranking remains a challenging and unanswered task.

To this end, we focus on the following questions:

---

*Work done during an internship at Baidu.
†Corresponding authors.

- **(RQ1)** How does ChatGPT perform on passage re-ranking tasks?
- **(RQ2)** How can we imitate the ranking capabilities of ChatGPT in a smaller, specialized model?

To answer the first question, we investigate prompting ChatGPT with two existing strategies (Sachan et al., 2022; Liang et al., 2022). However, we observe that they have limited performance and heavily rely on the availability of the log-probability of model output. Thus, we propose an alternative instructional **permutation generation** approach, instructing the LLMs to directly output the permutations of a group of passages. In addition, we propose an effective sliding window strategy to address context length limitations. For a comprehensive evaluation of LLMs, we employ three well-established IR benchmarks: TREC (Craswell et al., 2020), BEIR (Thakur et al., 2021), and My.TyDi (Zhang et al., 2021). Furthermore, to assess the LLMs on unknown knowledge and address concerns of data contamination, we suggest collecting a continuously updated evaluation testbed and propose **NovelEval**, a new test set with 21 novel questions.

To answer the second question, we introduce a **permutation distillation** technique to imitate the passage ranking capabilities of ChatGPT in a smaller, specialized ranking model. Specifically, we randomly sample 10K queries from the MS MARCO training set, and each query is retrieved by BM25 with 20 candidate passages. On this basis, we distill the permutation predicted by Chat-GPT into a student model using a RankNet-based distillation objective (Burges et al., 2005).

Our evaluation results demonstrate that GPT-4, equipped with zero-shot instructional permutation generation, surpasses supervised systems across nearly all datasets. Figure 1 illustrates that GPT-4 outperforms the previous state-of-the-art models by an average nDCG improvement of 2.7, 2.3, and 2.7 on TREC, BEIR, and My.TyDi, respectively. Furthermore, GPT-4 achieves state-of-the-art performance on the new NovelEval test set. Through our permutation distillation experiments, we observe that a 435M student model outperforms the previous state-of-the-art monoT5 (3B) model by an average nDCG improvement of 1.67 on BEIR. Additionally, the proposed distillation method demonstrates cost-efficiency benefits.

In summary, our contributions are tri-fold:

- We examine instructional methods for LLMs on passage re-ranking tasks and introduce a novel permutation generation approach; See Section 3 for details.
- We comprehensively evaluate ChatGPT and GPT-4 on various passage re-ranking benchmarks, including a newly proposed NovelEval test set; See Section 5 for details.
- We propose a distillation approach for learning specialized models with the permutation generated by ChatGPT; See Section 4 for details.

## 2 Related Work

### 2.1 Information Retrieval with LLMs

Recently, large language models (LLMs) have found increasing applications in information retrieval (Zhu et al., 2023). Several approaches have been proposed to utilize LLMs for passage retrieval. For example, SGPT (Muennighoff, 2022) generates text embeddings using GPT, generative document retrieval explores a differentiable search index (Tay et al., 2022; Cao et al., 2021; Sun et al., 2023), and HyDE (Gao et al., 2023; Wang et al., 2023a) generates pseudo-documents using GPT-3. In addition, LLMs have also been used for passage re-ranking tasks. UPR (Sachan et al., 2022) and SGPT-CE (Muennighoff, 2022) introduce instructional query generation methods, while HELM (Liang et al., 2022) utilizes instructional relevance generation. LLMs are also employed for training data generation. InPars (Bonifacio et al., 2022) generates pseudo-queries using GPT-3, and Promptagator (Dai et al., 2023) proposes a few-shot dense retrieval to leverage a few demonstrations from the target domain for pseudo-query generation. Furthermore, LLMs have been used for content generation (Yu et al., 2023) and web browsing (Nakano et al., 2021; Izacard et al., 2023; Microsoft, 2023). In this paper, we explore using ChatGPT and GPT-4 in passage re-ranking tasks, propose an instructional permutation generation method, and conduct a comprehensive evaluation of benchmarks from various domains, tasks, and languages. Recent work (Ma et al., 2023) concurrently investigated listwise passage re-ranking using LLMs. In comparison, our study provides a more comprehensive evaluation, incorporating a newly annotated dataset, and validates the proposed permutation distillation technique.

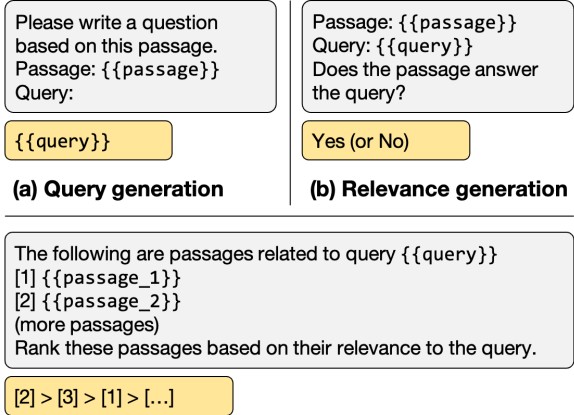

**(a) Query generation**

Please write a question based on this passage.
Passage: {{passage}}
Query:

{{query}}

**(b) Relevance generation**

Passage: {{passage}}
Query: {{query}}
Does the passage answer the query?

Yes (or No)

**(c) Permutation generation**

The following are passages related to query {{query}}
[1] {{passage_1}}
[2] {{passage_2}}
(more passages)
Rank these passages based on their relevance to the query.

[2] > [3] > [1] > [...]

Figure 2: Three types of instructions for zero-shot passage re-ranking with LLMs. The gray and yellow blocks indicate the inputs and outputs of the model. (a) Query generation relies on the log probability of LLMs to generate the query based on the passage. (b) Relevance generation instructs LLMs to output relevance judgments. (c) Permutation generation generates a ranked list of a group of passages. See Appendix A for details.

## 2.2 LLMs Specialization

Despite their impressive capabilities, LLMs such as GPT-4 often come with high costs and lack open-source availability. As a result, considerable research has explored ways to distill the capabilities of LLMs into specialized, custom models. For instance, Fu et al. (2023) and Magister et al. (2023) have successfully distilled the reasoning ability of LLMs into smaller models. Self-instruct (Wang et al., 2023b; Taori et al., 2023) propose iterative approaches to distill GPT-3 using their outputs. Additionally, Sachan et al. (2023) and Shi et al. (2023) utilize the generation probability of LLMs to improve retrieval systems. This paper presents a permutation distillation method that leverages Chat-GPT as a teacher to obtain specialized re-ranking models. Our experiments demonstrate that even with a small amount of ChatGPT-generated data, the specialized model can outperform strong supervised systems.

## 3 Passage Re-Ranking with LLMs

Ranking is the core task in information retrieval applications, such as ad-hoc search (Lin et al., 2020; Fan et al., 2022), Web search (Zou et al., 2021), and open-domain question answering (Karpukhin et al., 2020). Modern IR systems generally employ a multi-stage pipeline where the retrieval stage focuses on retrieving a set of candidates from a large

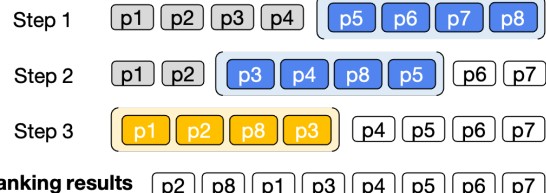

Step 1  p1 p2 p3 p4 [p5 p6 p7 p8]

Step 2  p1 p2 [p3 p4 p8 p5] p6 p7

Step 3  [p1 p2 p8 p3] p4 p5 p6 p7

Ranking results  p2 p8 p1 p3 p4 p5 p6 p7

Figure 3: Illustration of re-ranking 8 passages using sliding windows with a window size of 4 and a step size of 2. The blue color represents the first two windows, while the yellow color represents the last window. The sliding windows are applied in back-to-first order, meaning that the first 2 passages in the previous window will participate in re-ranking the next window.

corpus, and the re-ranking stage aims to re-rank this set to output a more precise list. Recent studies have explored LLMs for zero-shot re-ranking, such as instructional query generation or relevance generation (Sachan et al., 2022; Liang et al., 2022). However, existing methods have limited performance in re-ranking and heavily rely on the availability of the log probability of model output and thus cannot be applied to the latest LLMs such as GPT-4. Since ChatGPT and GPT-4 have a strong capacity for text understanding, instruction following, and reasoning, we introduce a novel **instructional permutation generation** method with a sliding window strategy to directly output a ranked list given a set of candidate passages. Figure 2 illustrates examples of three types of instructions; all the detailed instructions are included in Appendix A.

## 3.1 Instructional Permutation Generation

As illustrated in Figure 2 (c), our approach involves inputting a group of passages into the LLMs, each identified by a unique identifier (e.g., [1], [2], etc.). We then ask the LLMs to generate the permutation of passages in descending order based on their relevance to the query. The passages are ranked using the identifiers, in a format such as [2] > [3] > [1] > etc. The proposed method ranks passages directly without producing an intermediate relevance score.

## 3.2 Sliding Window Strategy

Due to the token limitations of LLMs, we can only rank a limited number of passages using the permutation generation approach. To overcome this constraint, we propose a sliding window strategy. Figure 3 illustrates an example of re-ranking 8 pas-

sages using a sliding window. Suppose the first-stage retrieval model returns $M$ passages. We re-rank these passages in a back-to-first order using a sliding window. This strategy involves two hyper-parameters: window size ($w$) and step size ($s$). We first use the LLMs to rank the passages from the $(M-w)$-th to the $M$-th. Then, we slide the window in steps of $s$ and re-rank the passages within the range from the $(M-w-s)$-th to the $(M-s)$-th. This process is repeated until all passages have been re-ranked.

## 4 Specialization by Permutation Distillation

Although ChatGPT and GPT-4 are highly capable, they are also too expensive to deploy in commercial search systems. Using GPT-4 to re-rank passages will greatly increase the latency of the search system. In addition, large language models suffer from the problem of unstable generation. Therefore, we argue that the capabilities of large language models are redundant for the re-ranking task. Thus, we can distill the re-ranking capability of large language models into a small model by specialization.

### 4.1 Permutation Distillation

In this paper, we present a novel permutation distillation method that aims to distill the passage re-ranking capability of ChatGPT into a specialized model. The key difference between our approach and previous distillation methods is that we directly use the model-generated permutation as the target, without introducing any inductive bias such as consistency-checking or log-probability manipulation (Bonifacio et al., 2022; Sachan et al., 2023). To achieve this, we sample 10,000 queries from MS MARCO and retrieve 20 candidate passages using BM25 for each query. The objective of distillation aims to reduce the differences between the permutation outputs of the student and ChatGPT.

### 4.2 Training Objective

Formally, suppose we have a query $q$ and $M$ passages $(p_1, \ldots, p_M)$ retrieved by BM25 ($M = 20$ in our implementation). ChatGPT with instructional permutation generation could produce the ranking results of the $M$ passages, denoted as $R = (r_1, \ldots, r_M)$, where $r_i \in [1, 2, \ldots, M]$ is the rank of the passage $p_i$. For example, $r_i = 3$ means $p_i$ ranks third among the $M$ passages. Now we have a specialized model $s_i = f_\theta(q, p_i)$ with

parameters $\theta$ to calculate the relevance score $s_i$ of paired $(q, p_i)$ using a cross-encoder. Using the permutation $R$ generated by ChatGPT, we consider RankNet loss (Burges et al., 2005) to optimize the student model:

$$\mathcal{L}_{\text{RankNet}} = \sum_{i=1}^{M} \sum_{j=1}^{M} \mathbb{1}_{r_i < r_j} \log(1 + \exp(s_i - s_j))$$

RankNet is a pairwise loss that measures the correctness of relative passage orders. When using permutations generated by ChatGPT, we can construct $M(M-1)/2$ pairs.

### 4.3 Specialized Model Architecture

Regarding the architecture of the specialized model, we consider two model structures: the BERT-like model and the GPT-like model.

#### 4.3.1 BERT-like model.

We utilize a cross-encoder model (Nogueira and Cho, 2019) based on DeBERTa-large. It concatenates the query and passage with a `[SEP]` token and estimates relevance using the representation of the `[CLS]` token.

#### 4.3.2 GPT-like model.

We utilize the LLaMA-7B (Touvron et al., 2023) with a zero-shot relevance generation instruction (see Appendix A). It classifies the query and passage as *relevance* or *irrelevance* by generating a relevance token. The relevance score is then defined as the generation probability of the relevance token.

Figure 5 illustrates the structure of the two types of specialized models.

## 5 Datasets

Our experiments are conducted on three benchmark datasets and one newly collected test set NovelEval.

### 5.1 Benchmark Datasets

The benchmark datasets include, TREC-DL (Craswell et al., 2020), BEIR (Thakur et al., 2021), and Mr.TyDi (Zhang et al., 2021).

**TREC** is a widely used benchmark dataset in IR research. We use the test sets of the 2019 and 2020 competitions: (i) TREC-DL19 contains 43 queries, (ii) TREC-DL20 contains 54 queries.

**BEIR** consists of diverse retrieval tasks and domains. We choose eight tasks in BEIR to evaluate the models: (i) *Covid*: retrieves scientific articles

for COVID-19 related questions. (ii) *NFCorpus* is a bio-medical IR data. (iii) *Touche* is an argument retrieval datasets. (iv) *DBPedia* retrieves entities from DBpedia corpus. (v) *SciFact* retrieves evidence for claims verification. (vi) *Signal* retrieves relevant tweets for a given news title. (vii) *News* retrieves relevant news articles for news headlines. (viii) *Robust04* evaluates poorly performing topics.

**Mr.TyDi** is a multilingual passages retrieval dataset of ten low-resource languages: Arabic, Bengali, Finnish, Indonesian, Japanese, Korean, Russian, Swahili, Telugu, and Thai. We use the first 100 samples in the test set of each language.

### 5.2 A New Test Set – NovelEval

The questions in the current benchmark dataset are typically gathered years ago, which raises the issue that existing LLMs already possess knowledge of these questions (Yu et al., 2023). Furthermore, since many LLMs do not disclose information about their training data, there is a potential risk of contamination of the existing benchmark test set (OpenAI, 2023). However, re-ranking models are expected to possess the capability to comprehend, deduce, and rank knowledge that is inherently unknown to them. Therefore, we suggest constructing continuously updated IR test sets to ensure that the questions, passages to be ranked, and relevance annotations have not been learned by the latest LLMs for a fair evaluation.

As an initial effort, we built **NovelEval-2306**, a novel test set with 21 novel questions. This test set is constructed by gathering questions and passages from 4 domains that were published after the release of GPT-4. To ensure that GPT-4 did not possess prior knowledge of these questions, we presented them to both `gpt-4-0314` and `gpt-4-0613`. For instance, question *"Which film was the 2023 Palme d'Or winner?"* pertains to the Cannes Film Festival that took place on May 27, 2023, rendering its answer inaccessible to most existing LLMs. Next, we searched 20 candidate passages for each question using Google search. The relevance of these passages was manually labeled as: 0 for not relevant, 1 for partially relevant, and 2 for relevant. See Appendix C for more details.

## 6 Experimental Results of LLMs

### 6.1 Implementation and Metrics

In benchmark datasets, we re-rank the top-100 passages retrieved by BM25 using pyserini[1] and use nDCG@{1, 5,10} as evaluation metrics. Since ChatGPT cannot manage 100 passages at a time, we use the sliding window strategy introduced in Section 3.2 with a window size of 20 and step size of 10. In NovelEval, we randomly shuffled the 20 candidate passages searched by Google and re-ranked them using ChatGPT and GPT-4 with permutation generation.

### 6.2 Results on Benchmarks

On benchmarks, we compare ChatGPT and GPT-4 with state-of-the-art supervised and unsupervised passage re-ranking methods. The supervised baselines include: monoBERT (Nogueira and Cho, 2019), monoT5 (Nogueira et al., 2020), mmarcoCE (Bonifacio et al., 2021), and Cohere Rerank[2]. The unsupervised baselines include: UPR (Sachan et al., 2022), InPars (Bonifacio et al., 2022), and Promptagator++ (Dai et al., 2023). See Appendix E for more details on implementing the baseline.

Table 1 presents the evaluation results obtained from the TREC and BEIR datasets. The following observations can be made: (i) GPT-4, when equipped with the permutation generation instruction, demonstrates superior performance on both datasets. Notably, GPT-4 achieves an average improvement of 2.7 and 2.3 in nDCG@10 on TREC and BEIR, respectively, compared to monoT5 (3B). (ii) ChatGPT also exhibits impressive results on the BEIR dataset, surpassing the majority of supervised baselines. (iii) On BEIR, we use only GPT-4 to re-rank the top-30 passages re-ranked by Chat-GPT. The method achieves good results, while the cost is only 1/5 of that of only using GPT-4 for re-ranking.

Table 2 illustrates the results on Mr. TyDi of ten low-resource languages. Overall, GPT-4 outperforms the supervised system in most languages, demonstrating an average improvement of 2.65 nDCG over mmarcoCE. However, there are instances where GPT-4 performs worse than mmarcoCE, particularly in low-resource languages like Bengali, Telugu, and Thai. This may be attributed to the weaker language modeling ability of GPT-4

---

[1] https://github.com/castorini/pyserini
[2] https://txt.cohere.com/rerank/

| Method | DL19 | DL20 | Covid | NFCorpus | Touche | DBPedia | SciFact | Signal | News | Robust04 | BEIR (Avg) |
|---|---|---|---|---|---|---|---|---|---|---|---|
| BM25 | 50.58 | 47.96 | 59.47 | 30.75 | **44.22** | 31.80 | 67.89 | 33.05 | 39.52 | 40.70 | 43.42 |
| **Supervised** | | | | | | | | | | | |
| monoBERT (340M) | 70.50 | 67.28 | 70.01 | 36.88 | 31.75 | 41.87 | 71.36 | 31.44 | 44.62 | 49.35 | 47.16 |
| monoT5 (220M) | 71.48 | 66.99 | 78.34 | 37.38 | 30.82 | 42.42 | 73.40 | 31.67 | 46.83 | 51.72 | 49.07 |
| monoT5 (3B) | 71.83 | 68.89 | 80.71 | **38.97** | 32.41 | 44.45 | **76.57** | 32.55 | 48.49 | 56.71 | 51.36 |
| Cohere Rerank-v2 | 73.22 | 67.08 | 81.81 | 36.36 | 32.51 | 42.51 | 74.44 | 29.60 | 47.59 | 50.78 | 49.45 |
| **Unsupervised** | | | | | | | | | | | |
| UPR (FLAN-T5-XL) | 53.85 | 56.02 | 68.11 | 35.04 | 19.69 | 30.91 | 72.69 | 31.91 | 43.11 | 42.43 | 42.99 |
| InPars (monoT5-3B) | - | 66.12 | 78.35 | - | - | - | - | - | - | - | - |
| Promptagator++ (few-shot) | - | - | 76.2 | 37.0 | 38.1 | 43.4 | 73.1 | - | - | - | - |
| **LLM API (Permutation generation)** | | | | | | | | | | | |
| gpt-3.5-turbo | 65.80 | 62.91 | 76.67 | 35.62 | 36.18 | 44.47 | 70.43 | 32.12 | 48.85 | 50.62 | 49.37 |
| gpt-4[†] | **75.59** | **70.56** | **85.51** | **38.47** | 38.57 | 47.12 | 74.95 | **34.40** | **52.89** | **57.55** | **53.68** |

Table 1: **Results (nDCG@10) on TREC and BEIR.** Best performing unsupervised and overall system(s) are marked bold. All models except InPars and Promptagator++ re-rank the same BM25 top-100 passages. [†]On BEIR, we use gpt-4 to re-rank the top-30 passages re-ranked by gpt-3.5-turbo to reduce the cost of calling gpt-4 API.

| Method | BM25 | mmarcoCE | gpt-3.5 | +gpt-4 |
|---|---|---|---|---|
| Arabic | 39.19 | 68.18 | 71.00 | **72.56** |
| Bengali | 45.56 | **65.98** | 53.10 | 64.37 |
| Finnish | 29.91 | 54.15 | 56.48 | **62.29** |
| Indonesian | 51.79 | 69.94 | 68.45 | **75.47** |
| Japanese | 27.39 | 49.80 | 50.70 | **58.22** |
| Korean | 26.29 | 44.00 | 41.48 | **49.63** |
| Russian | 34.04 | 53.16 | 48.75 | **53.45** |
| Swahili | 45.15 | 60.31 | 62.38 | **67.67** |
| Telugu | 37.05 | **68.92** | 51.69 | 62.22 |
| Thai | 44.62 | **68.36** | 55.57 | 63.41 |
| Avg | 38.10 | 60.28 | 55.96 | **62.93** |

Table 2: Results (nDCG@10) on Mr.TyDi.

| Method | nDCG@1 | nDCG@5 | nDCG@10 |
|---|---|---|---|
| BM25 | 33.33 | 45.96 | 55.77 |
| monoBERT (340M) | 78.57 | 70.65 | 77.27 |
| monoT5 (220M) | 83.33 | 77.46 | 81.27 |
| monoT5 (3B) | 83.33 | 78.38 | 84.62 |
| gpt-3.5-turbo | 76.19 | 74.15 | 75.71 |
| gpt-4 | **85.71** | **87.49** | **90.45** |

Table 3: Results on NovelEval.

| Method | DL19 nDCG@1/5/10 | DL20 nDCG@1/5/10 |
|---|---|---|
| curie-001 | RG 39.53 / 40.02 / 41.53 | 41.98 / 34.80 / 34.91 |
| curie-001 | QG 50.78 / 50.77 / 49.76 | 50.00 / 48.36 / 48.73 |
| curie-001 | PG 66.67 / 56.79 / 54.21 | 59.57 / 55.20 / 52.17 |
| davinci-003 | RG 54.26 / 52.78 / 50.58 | 64.20 / 58.41 / 56.87 |
| davinci-003 | QG 37.60 / 44.73 / 45.37 | 51.25 / 47.46 / 45.93 |
| davinci-003 | PG 69.77 / 64.73 / 61.50 | 69.75 / 58.76 / 57.05 |
| gpt-3.5 | PG 82.17 / 71.15 / 65.80 | **79.32** / 66.76 / 62.91 |
| gpt-4 | PG **82.56** / **79.16** / **75.59** | 78.40 / **74.11** / **70.56** |

Table 4: **Compare different instruction and API endpoint.** Best performing system(s) are marked bold. PG, QG, RG denote permutation generation, query generation and relevance generation, respectively.

in these languages and the fact that text in low-resource languages tends to consume more tokens than English text, leading to the over-cropping of passages. Similar trends are observed with Chat-GPT, which is on par with the supervised system in most languages, and consistently trails behind GPT-4 in all languages.

## 6.3 Results on NovelEval

Table 3 illustrates the evaluation results on our newly collected NovelEval, a test set containing 21 novel questions and 420 passages that GPT-4 had not learned. The results show that GPT-4 performs well on these questions, significantly outperforming the previous best-supervised method, monoT5 (3B). Additionally, ChatGPT achieves a performance level comparable to that of monoBERT. This outcome implies that LLMs possess the capability to effectively re-rank unfamiliar information.

## 6.4 Compare with Different Instructions

We conduct a comparison with the proposed permutation generation (PG) with previous query generation (QG) (Sachan et al., 2022) and relevance generation (RG) (Liang et al., 2022) on TREC-DL19. An example of the three types of instructions is in Figure 2, and the detailed implementation is in Appendix B. We also compare four LLMs provided

| | Method | nDCG@1 | nDCG@5 | nDCG@10 |
|---|---|---|---|---|
| | BM25 | 54.26 | 52.78 | 50.58 |
| | gpt-3.5-turbo | 82.17 | 71.15 | 65.80 |
| | *Initial passage order* | | | |
| (1) | Random order | 26.36 | 25.32 | 25.17 |
| (2) | Reverse order | 36.43 | 31.79 | 32.77 |
| | *Number of re-ranking* | | | |
| (3) | Re-rank 2 times | 78.29 | 69.37 | 66.62 |
| (4) | Re-rank 3 times | 78.29 | 69.74 | 66.97 |
| (5) | gpt-4 Rerank | 80.23 | 76.70 | 73.64 |

Table 5: **Ablation study on TREC-DL19.** We use gpt-3.5-turbo with permutation generation with different configuration.

in the OpenAI API[3]: curie-001 - GPT-3 model with about 6.7 billion parameters (Brown et al., 2020); davinci-003 - GPT-3.5 model trained with RLHF and about 175 billion parameters (Ouyang et al., 2022); gpt-3.5-turbo - The underlying model of ChatGPT (OpenAI, 2022); gpt-4 - GPT-4 model (OpenAI, 2023).

The results are listed in Table 4. From the results, we can see that: (i) The proposed PG method outperforms both QG and RG methods in instructing LLMs to re-rank passages. We suggest two explanations: First, from the result that PG has significantly higher top-1 accuracy compared to other methods, we infer that LLMs can explicitly compare multiple passages with PG, allowing subtle differences between passages to be discerned. Second, LLMs gain a more comprehensive understanding of the query and passages by reading multiple passages with potentially complementary information, thus improving the model's ranking ability. (ii) With PG, ChatGPT performs comparably to GPT-4 on nDCG@1, but lags behind it on nDCG@10. The Davinci model (text-davinci-003) performs poorly compared to ChatGPT and GPT-4. This may be because of the generation stability of Davinci and ChatGPT trails that of GPT-4. We delve into the stability analysis of Davinci, ChatGPT, and GPT-4 in Appendix F.

### 6.5 Ablation Study on TREC

We conducted an ablation study on TREC to gain insights into the detailed configuration of permutation generation. Table 5 illustrates the results.

[3]https://platform.openai.com/docs/api-reference

**Initial Passage Order** While our standard implementation utilizes the ranking result of BM25 as the initial order, we examined two alternative variants: random order (1) and reversed BM25 order (2). The results reveal that the model's performance is highly sensitive to the initial passage order. This could be because BM25 provides a relatively good starting passage order, enabling satisfactory results with only a single sliding window re-ranking.

**Number of Re-Ranking** Furthermore, we studied the influence of the number of sliding window passes. Models (3-4) in Table 5 show that re-ranking more times may improve nDCG@10, but it somehow hurts the ranking performance on top passages (e.g., nDCG@1 decreased by 3.88). Re-ranking the top 30 passages using GPT-4 showed notable accuracy improvements (see the model (5)). This provides an alternative method to combine ChatGPT and GPT-4 in passage re-ranking to reduce the high cost of using the GPT-4 model.

### 6.6 Results of LLMs beyond ChatGPT

We further test the capabilities of other LLMs beyond the OpenAI series on TREC DL-19. As shown in Table 6, we evaluate the top-20 BM25 passage re-ranking nDCG of proprietary LLMs from OpenAI, Cohere, Antropic, and Google, and three open-source LLMs. We see that: (i) Among the proprietary LLMs, GPT-4 exhibited the highest re-ranking performance. Cohere Re-rank also showed promising results; however, it should be noted that it is a supervised specialized model. In contrast, the proprietary models from Antropic and Google fell behind ChatGPT in terms of re-ranking effectiveness. (ii) As for the open-source LLMs, we observed a significant performance gap compared to ChatGPT. One possible reason for this discrepancy could be the complexity involved in generating permutations of 20 passages, which seems to pose a challenge for the existing open-source models.

We analyze the model's unexpected behavior in Appendix F, and the cost of API in Appendix H.

## 7 Experimental Results of Specialization

As mentioned in Section 4, we randomly sample 10K queries from the MSMARCO training set and employ the proposed permutation distillation to distill ChatGPT's predicted permutation into specialized re-ranking models. The specialized re-ranking models could be DeBERTa-v3-Large with a cross-encoder architecture or LLaMA-7B with relevance

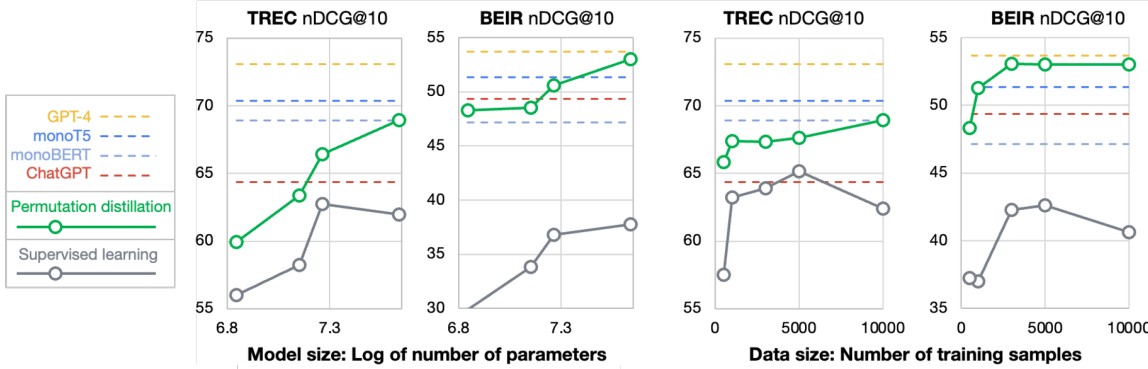

Figure 4: **Scaling experiment.** The dashed line indicates the baseline methods: GPT-4, monoT5, monoBERT, and ChatGPT. The solid green line and solid gray line indicate the specialized Deberta models obtained by the proposed *permutation distillation* and by *supervised learning* on MS MARCO, respectively. This figure compares the models' performance on TREC and BEIR across varying model sizes (70M to 435M) and training data sizes (500 to 10K).

| Method | ND1 | ND5 | ND10 |
|---|---|---|---|
| OpenAI `text-davinci-003` | 70.54 | 61.90 | 57.24 |
| OpenAI `gpt-3.5-turbo` | 75.58 | 66.19 | 60.89 |
| OpenAI `gpt-4` | 79.46 | **71.65** | **65.68** |
| Cohere `rerank-english-v2.0` | 79.46 | 71.56 | 64.78 |
| Antropic `claude-2` | 66.66 | 59.33 | 55.91 |
| Antropic `claude-instant-1` | **81.01** | 66.71 | 62.23 |
| Google `text-bison-001` | 69.77 | 64.46 | 58.67 |
| Google `bard-2023.10.21` | **81.01** | 65.57 | 60.11 |
| Google `flan-t5-xxl` | 52.71 | 51.63 | 50.26 |
| Tsinghua `ChatGLM-6B` | 54.26 | 52.77 | 50.58 |
| LMSYS `Vicuna-13B` | 54.26 | 51.55 | 49.08 |

Table 6: **Results of different LLMs on re-ranking top-20 passages on DL-19.** ND{1,5,10} denote nDCG@{1,5,10}, respectively.

| Label | Method | DL19 | DL20 | BEIR (Avg) |
|---|---|---|---|---|
| ∅ | BM25 | 50.58 | 47.96 | 43.42 |
| ∅ | ChatGPT | 65.80 | 62.91 | 49.37 |
| MARCO | monoT5 (3B) | 71.83 | 68.89 | 51.36 |
| MARCO | DeBERTa-Large | 68.89 | 61.38 | 42.64 |
| MARCO | LLaMA-7B | 69.24 | 58.97 | 47.71 |
| ChatGPT | DeBERTa-Large | 70.66 | **67.15** | **53.03** |
| ChatGPT | LLaMA-7B | **71.78** | 66.89 | 51.68 |

Table 7: **Results (nDCG@10) of specialized models.** Best performing specialized model(s) are marked bold. The label column denotes the relevance judgements used in model training, where MARCO denotes use MS MARCO annotation, ChatGPT denotes use the outputs of permutation generation instructed ChatGPT as labels. BEIR (Avg) denotes average nDCG on eight BEIR datasets, and the detailed results are at Table 13.

generation instructions. We also implemented the specialized model trained using the original MS MARCO labels (aka supervised learning) for comparison[4].

## 7.1 Results on Benchmarks

Table 7 lists the results of specialized models, and Table 13 includes the detailed results. Our findings can be summarized as follows: (i) Permutation distillation outperforms the supervised counterpart on both TREC and BEIR datasets, potentially because ChatGPT's relevance judgments are more comprehensive than MS MARCO labels (Arabzadeh et al., 2021). (ii) The specialized DeBERTa model outperforms previous state-of-the-art (SOTA) baselines, monoT5 (3B), on BEIR with an average nDCG of 53.03. This result highlights the potential of distilling LLMs for IR since it is significantly more cost-efficient. (iii) The distilled specialized model also surpasses ChatGPT, its teacher model, on both datasets. This is probably because the re-ranking stability of specialized models is better than ChatGPT. As shown in the stability analysis in Appendix F, ChatGPT is very unstable in generating permutations.

## 7.2 Analysis on Model Size and Data Size

In Figure 4, we present the re-ranking performance of specialized DeBERTa models obtained through permutation distillation and supervised learning of different model sizes (ranging from 70M to 435M) and training data sizes (ranging from 500 to 10K). Our findings indicate that the permutation-distilled models consistently outperform their supervised counterparts across all settings, particularly on the BEIR datasets. Notably, even with only 1K training queries, the permutation-distilled DeBERTa model

---
[4]Note that all models are trained using the RankNet loss for a fair comparison.

achieves superior performance compared to the previous state-of-the-art monoT5 (3B) model on BEIR. We also observe that increasing the number of model parameters yields a greater improvement in the ranking results than increasing the training data. Finally, we find that the performance of supervised models is unstable for different model sizes and data sizes. This may be due to the presence of noise in the MS MARCO labels, which leads to overfitting problems (Arabzadeh et al., 2021).

# 8 Conclusion

In this paper, we conduct a comprehensive study on passage re-ranking with LLMs. We introduce a novel permutation generation approach to fully explore the power of LLMs. Our experiments on three benchmarks have demonstrated the capability of ChatGPT and GPT-4 in passage re-ranking. To further validate LLMs on unfamiliar knowledge, we introduce a new test set called NovelEval. Additionally, we propose a permutation distillation method, which demonstrates superior effectiveness and efficiency compared to existing supervised approaches.

## Limitations

The limitations of this work include the main analysis for OpenAI ChatGPT and GPT-4, which are proprietary models that are not open-source. Although we also tested on open-source models such as FLAN-T5, ChatGLM-6B, and Vicuna-13B, the results still differ significantly from ChatGPT. How to further exploit the open-source models is a question worth exploring. Additionally, this study solely focuses on examining LLMs in the re-ranking task. Consequently, the upper bound of the ranking effect is contingent upon the recall of the initial passage retrieval. Our findings also indicate that the re-ranking effect of LLMs is highly sensitive to the initial order of passages, which is usually determined by the first-stage retrieval, such as BM25. Therefore, there is a need for further exploration into effectively utilizing LLMs to enhance the first-stage retrieval and improve the robustness of LLMs in relation to the initial passage retrieval.

## Ethics Statement

We acknowledge the importance of the ACM Code of Ethics and totally agree with it. We ensure that this work is compatible with the provided code, in terms of publicly accessed datasets and models. Risks and harms of large language models include the generation of harmful, offensive, or biased content. These models are often prone to generating incorrect information, sometimes referred to as hallucinations. We do not expect the studied model to be an exception in this regard. The LLMs used in this paper were shown to suffer from bias, hallucination, and other problems. Therefore, we are not recommending the use of LLMs for ranking tasks with social implications, such as ranking job candidates or ranking products, because LLMs may exhibit racial bias, geographical bias, gender bias, etc., in the ranking results. In addition, the use of LLMs in critical decision-making sessions may pose unspecified risks. Finally, the distilled models are licensed under the terms of OpenAI because they use ChatGPT. The distilled LLaMA models are further licensed under the non-commercial license of LLaMA.

## Acknowledgements

This work was supported by the Natural Science Foundation of China (62272274, 61972234, 62072279, 62102234, 62202271), the Natural Science Foundation of Shandong Province (ZR2021QF129, ZR2022QF004), the Key Scientific and Technological Innovation Program of Shandong Province (2019JZZY010129), the Fundamental Research Funds of Shandong University, the China Scholarship Council under grant nr. 202206220085.

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

# A  Instructions

## A.1  Query Generation Instruction

The query generation instruction (Sachan et al., 2022) uses the log-probability of the query.

> Please write a question based on this passage.
> Passage: {{passage}}
> Question: {{query}}

## A.2  Relevance Generation Instruction (few-shot)

Following HELM (Liang et al., 2022), the relevance generation instruction use 4 in-context examples.

> Given a passage and a query, predict whether the passage includes an answer to the query by producing either 'Yes' or 'No'.
>
> Passage: Its 25 drops per ml, you guys are all wrong. If it is water, the standard was changed 15 - 20 years ago to make 20 drops = 1mL. The viscosity of most things is temperature dependent, so this would be at room temperature. Hope this helps.
> Query: how many eye drops per ml
> Does the passage answer the query?
> Answer: Yes
>
> Passage: RE: How many eyedrops are there in a 10 ml bottle of Cosopt? My Kaiser pharmacy insists that 2 bottles should last me 100 days but I run out way before that time when I am using 4 drops per day.In the past other pharmacies have given me 3 10-ml bottles for 100 days.E: How many eyedrops are there in a 10 ml bottle of Cosopt? My Kaiser pharmacy insists that 2 bottles should last me 100 days but I run out way before that time when I am using 4 drops per day.
> Query: how many eye drops per ml
> Does the passage answer the query?
> Answer: No
>
> Passage: : You can transfer money to your checking account from other Wells Fargo. accounts through Wells Fargo Mobile Banking with the mobile app, online, at any. Wells Fargo ATM, or at a Wells Fargo branch. 1 Money in — deposits.
> Query: can you open a wells fargo account online
> Does the passage answer the query?
> Answer: No
>
> Passage: You can open a Wells Fargo banking account from your home or even online. It is really easy to do, provided you have all of the appropriate documentation. Wells Fargo has so many bank account options that you will be sure to find one that works for you. They offer free checking accounts with free online banking.
> Query: can you open a wells fargo account online
> Does the passage answer the query?
> Answer: Yes
>
> Passage: {{passage}}
> Query:{{query}}
> Does the passage answer the query?
> Answer:

## A.3 Relevance Generation Instruction (zero-shot)

This instruction is used to train LLaMA-7B specialized models.

> Given a passage and a query, predict whether the passage includes an answer to the query by producing either 'Yes' or 'No'.
>
> Passage: {{passage}}
> Query: {{query}}
> Does the passage answer the query?
> Answer:

## A.4 Permutation Generation Instruction (Text)

Permutation generation (text) is used for `text-davinci-003`.

> This is RankGPT, an intelligent assistant that can rank passages based on their relevancy to the query.
>
> The following are {{num}} passages, each indicated by number identifier []. I can rank them based on their relevance to query: {{query}}
>
> [1] {{passage_1}}
>
> [2] {{passage_2}}
>
> (more passages) ...
>
> The search query is: {{query}}
>
> I will rank the {{num}} passages above based on their relevance to the search query. The passages will be listed in descending order using identifiers, and the most relevant passages should be listed first, and the output format should be [] > [] > etc, e.g., [1] > [2] > etc.
>
> The ranking results of the {{num}} passages (only identifiers) is:

## A.5 Permutation Generation Instruction (Chat)

Permutation generation instruction (chat) is used for `gpt-3.5-turbo` and `gpt-4`.

> **system:**
> You are RankGPT, an intelligent assistant that can rank passages based on their relevancy to the query.
>
> **user:**
> I will provide you with {{num}} passages, each indicated by number identifier []. Rank them based on their relevance to query: {{query}}.
>
> **assistant:**
> Okay, please provide the passages.
>
> **user:**
> [1] {{passage_1}}
>
> **assistant:**
> Received passage [1]
>
> **user:**
> [2] {{passage_2}}
>
> **assistant:**
> Received passage [2]
>
> (more passages) ...
>
> **user**
> Search Query: {{query}}.
> Rank the {{num}} passages above based on their relevance to the search query. The passages should be listed in descending order using identifiers, and the most relevant passages should be listed first, and the output format should be [] > [], e.g., [1] > [2]. Only response the ranking results, do not say any word or explain.

## B  Instructional Methods on LLMs as Rernaker

This paper focus on re-ranking task, given $M$ passages for a query $q$, the re-ranking aims to use an agent $f(\cdot)$ to output their ranking results $\mathbf{R} = (r_1, ..., r_M)$, where $r_i \in [1, 2, ..., M]$ denotes the rank of $p_i$. This paper studies using the LLMs as $f(\cdot)$.

### B.1  Instructional Query Generation

Query generation has been studied in Sachan et al. (2022); Muennighoff (2022), in which the relevance between a query and a passage is measured by the log-probability of the model to generate the query based on the passage. Figure 2 (a) shows an example of instructional query generation.

Formally, given query $q$ and a passage $p_i$, their relevance score $s_i$ is calculated as:

$$s_i = \frac{1}{|q|} \sum_t \log p(q_t | q_{<t}, p_i, \mathcal{I}_{\text{query}}) \tag{1}$$

where $|q|$ denotes the number of tokens in $q$, $q_t$ denotes the $t$-th token of $q$, and $\mathcal{I}_{\text{query}}$ denotes the instructions, referring to Figure 2 (a). The passages are then ranked based on relevance score $s_i$.

### B.2  Instructional Relevance Generation

Relevance generation is employed in HELM (Liang et al., 2022). Figure 2 (b) shows an example of instructional relevance generation, in which LLMs are instructed to output "Yes" if the query and passage

are relevant or "No" if they are irrelevant. The relevance score $s_i$ is measured by the probability of LLMs generating the word 'Yes' or 'No':

$$s_i = \begin{cases} 1 + p(\text{Yes}), & \text{if output is Yes} \\ 1 - p(\text{No}), & \text{if output is No} \end{cases} \qquad (2)$$

where $p(\text{Yes/No})$ denotes the probability of LLMs generating Yes or No, and the relevance score is normalized into the range [0, 2].

The above two methods rely on the log probability of LLM, which is often unavailable for LLM API. For example, at the time of writing, OpenAI's `ChatCompletion` API does not provide the log-probability of generation[5].

### B.3 Instructional Permutation Generation

The proposed instructional permutation generation is a listwise approach, which directly assigns each passage $p_i$ a unique ranking identifier $a_i$ (e.g., [1], [2]) and places it at the beginning of $p_i$: $p'_i = \text{Concat}(a_i, p_i)$. Subsequently, a generative LLM is instructed to generate a permutation of these identifiers: **Perm** $= f(q, p'_1, ..., p'_M)$, where the permutation **Perm** indicates the rank of the identifiers $a_i$ (e.g., [1], [2]). We then simply map the identifiers $a_i$ to the passages $p_i$ to obtain the ranking of the passages.

| Domain | Question | Reference Answer |
|--------|----------|------------------|
| Sport | What is Messi's annual income after transferring to Miami? | $50M-$60M |
| Sport | How many goals did Haaland scored in the 2023 Champions League Final? | 0 |
| Sport | Where did Benzema go after leaving Real Madrid? | Saudi Arabia |
| Sport | Where was the 2023 Premier League FA Cup Final held? | Wembley Stadium |
| Sport | Who won 2023 Laureus World Sportsman Of The Year Award? | Lionel Messi |
| Sport | Who wins NBA Finals 2023? | Denver Nuggets |
| Tech | What is the screen resolution of vision pro? | 4K with one eye |
| Tech | What is the name of the combined Deepmind and Google Brain? | Google DeepMind |
| Tech | How much video memory does the DGX GH200 have? | 144TB |
| Tech | What are the new features of PyTorch 2? | faster, low memory, dynamic shapes |
| Tech | Who will be the CEO of Twitter after Elon Musk is no longer the CEO? | Linda Yaccarino |
| Tech | What are the best papers of CVPR 2023? | Visual Programming: Compositional [...] |
| Movie | Who sang the theme song of Transformers Rise of the Beasts? | Notorious B.I.G |
| Movie | Who is the villain in The Flash? | Eobard Thawne/Professor Zoom |
| Movie | How many different Spider-Men are there in Across the Spider-Verse? | 280 variations |
| Movie | Who does Momoa play in Fast X? | Dante |
| Movie | The Little Mermaid first week box office? | $163.8 million worldwide |
| Movie | Which film was the 2023 Palme d'Or winner? | Anatomy of a Fall |
| Other | Where will Blackpink's 2023 world tour concert in France be held? | the Stade de France |
| Other | What is the release date of song Middle Ground? | May 19, 2023 |
| Other | Where did the G7 Summit 2023 take place? | Hiroshima |

Table 8: Questions and reference answers on NovelEval-2306.

## C  NovelEval-2306

Table 8 lists the collected 21 questions. These questions come from four domains and include hot topics from the past few months. For each question, we used Google search to obtain 20 passages. When using Google search, in order to avoid all pages containing the answer, we used not only the question itself as a search query, but also the entities that appear in the question as an alternative search query to obtain some pages that are relevant but do not contain the answer. For example, for the first question *"What is Messi's annual income after transferring to Miami?"*, we used *"Messi"* and *"Messi transferring"* as search queries to get some pages that do not contain the answer. When searching, we collected the highest-ranking web pages, news, and used a paragraph or paragraphs from the web pages related to the search term as candidate passages. Table 9 shows the statistical information of the data. All of the LLMs (including

---

[5] https://platform.openai.com/docs/api-reference/chat/create

`gpt-4-0314` and `gpt-4-0613`) we tested achieved 0% question-answering accuracy on the obtained test set.

We searched for 20 candidate passages for each question using Google search. These passages were manually labeled for relevance by a group of annotators, including the authors and their highly educated colleagues. To ensure consistency, the annotation process was repeated twice. Each passage was assigned a relevance score: 0 for not relevant, 1 for partially relevant, and 2 for relevant. When evaluating the latest LLMs, we found that all non-retrieval-augmented models tested achieved 0% accuracy in answering the questions on the test set. This test set provides a reasonable evaluation of the latest LLMs at the moment. Since LLMs may be continuously trained on new data, the proposed test set should be continuously updated to counteract the contamination of the test set by LLMs.

| | |
|---|---|
| Number of questions | 21 |
| Number of passages | 420 |
| Number of relevance annotation | 420 |
| Average number words of passage | 149 |
| Number of score 0 | 290 |
| Number of score 1 | 40 |
| Number of score 2 | 90 |

Table 9: Data Statistics of NovelEval.

## D  Implementation Details

### D.1  Training Configuration

We use DeBERTa-V3-base, which concatenates the query and passage with a `[SEP]` token and utilizes the representation of the `[CLS]` token. To generate candidate passages, we randomly sample 10k queries and use BM25 to retrieve 20 passages for each query. We then re-rank the candidate passages using the `gpt-3.5-turbo` API with permutation generation instructions, at a cost of approximately $40. During training, we employ a batch size of 32 and utilize the AdamW optimizer with a constant learning rate of $5 \times 10^{-5}$. The model is trained for two epochs. Additionally, we implement models using the original MS MARCO labels for comparison.

The LLaMA-7B model is optimized with the AdamW optimizer, a constant learning rate of $5 \times 10^{-5}$, and with mixed precision of bf16 and Deepspeed Zero3 strategy. All the experiments are conducted on 8 A100-40G GPUs.

Figure 5 illustrates the detailed model architecture of BERT-like and GPT-like specialized models.

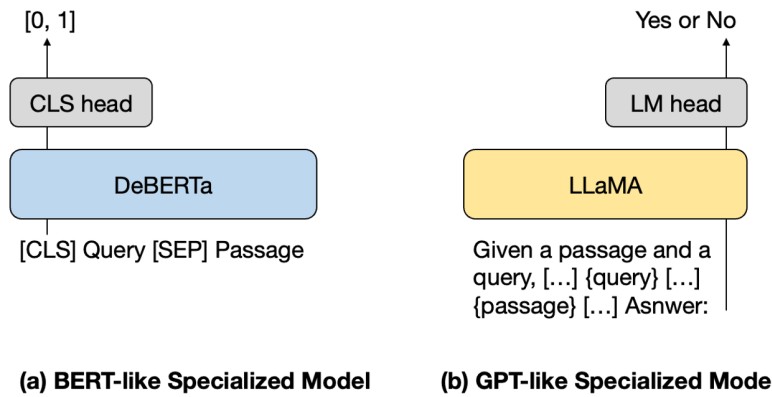

(a) BERT-like Specialized Model   (b) GPT-like Specialized Model

Figure 5: Model architecture of BERT-like and GPT-like specialized models.

## D.2 Training Objective

Using the permutation generated by ChatGPT, we consider the following losses to optimize the student model:

**Listwise Cross-Entropy (CE)** (Bruch et al., 2019). Listwise CE is the wide-use loss for passage ranking, which considers only one positive passage and defines the list-wise softmax cross-entropy on all candidate's passages:

$$\mathcal{L}_{\text{Listwise\_CE}} = -\sum_{i=1}^{M} \mathbb{1}_{r_i=1} \log(\frac{\exp(s_i)}{\sum_j \exp(s_j)})$$

where $\mathbb{1}$ is the indicator function.

**RankNet** (Burges et al., 2005). RankNet is a pairwise loss that measures the correctness of relative passage orders:

$$\mathcal{L}_{\text{RankNet}} = \sum_{i=1}^{M} \sum_{j=1}^{M} \mathbb{1}_{r_i < r_j} \log(1 + \exp(s_i - s_j))$$

when using permutation generated by ChatGPT, we can construct $M(M-1)/2$ pairs.

**LambdaLoss** (Wang et al., 2018). The LambdaLoss further accounts for the nDCG gains of the model ranks. LambdaLoss uses the student model's rank, denoted as $\pi = (\pi_1, \ldots, \pi_M)$, where $\pi_i$ is the model predicted rank of $p_i$ with a similar definition with ChatGPT rank $R$. The loss function is defined as:

$$\mathcal{L}_{\text{Lambda}} = \sum_{r_i < r_j} \Delta\text{NDCG} \log_2(1 + \exp(s_i - s_j))$$

in which $\Delta\text{NDCG}$ is the delta of NDCG which could be compute as $\Delta\text{NDCG} = |G_i - G_j||\frac{1}{D(\pi_i)} - \frac{1}{D(\pi_j)}|$, where $D(\pi_i)$ and $D(\pi_j)$ are the position discount functions and $G_i$ and $G_j$ are the gain functions used in NDCG (Wang et al., 2018).

**Pointwise Binary Cross-Entropy (BCE).** We also include the Pointwise BCE as the baseline loss for supervised methods, which is calculated based on each query-document pair independently:

$$\mathcal{L}_{\text{BCE}} = -\sum_{i=1}^{M} \mathbb{1}_{r_i=1} \log \sigma(s_i) + \mathbb{1}_{r_i \neq 1} \log \sigma(1 - s_i)$$

where $\sigma(x) = \frac{1}{1+\exp(-x)}$ is the logistic function.

## E    Baselines Details

We include state-of-the-art supervised and unsupervised passage re-ranking methods for comparison. The **supervised baselines** are:

- **monoBERT** (Nogueira and Cho, 2019): A cross-encoder re-ranker based on BERT-large, trained on MS MARCO.
- **monoT5** (Nogueira et al., 2020): A sequence-to-sequence re-ranker that uses T5 to calculate the relevance score[6].
- **Cohere Rerank**: A passage reranking API `rerank-english-v2.0` developed by Cohere[7]. Cohere does not provide details on the structure and training method of the model.
- **mmarcoCE** (Bonifacio et al., 2021): A 12-layer mMiniLM-v2 cross-encoder model[8] trained on mmarco, a translated version of MS MARCO. mmarcoCE serves as a baseline for Mr.TyDi.

The **unsupervised baselines** are:

---

[6]https://huggingface.co/castorini/monot5-3b-msmarco-10k
[7]https://cohere.com/rerank
[8]https://huggingface.co/cross-encoder/mmarco-mMiniLMv2-L12-H384-v1

- **UPR** (Sachan et al., 2022): Unsupervised passage ranking with instructional query generation. Due to its superior performance, we use the FLAN-T5-XL (Chung et al., 2022) as the LLM of UPR.
- **InPars** (Bonifacio et al., 2022): monoT5-3B trained on pseudo data generated by GPT-3.
- **Promptagator++** (Dai et al., 2023): A 110M cross-encoder re-ranker trained on pseudo queries generated by FALN 137B.

| Method | Repetition↓ | Missing↓ | Rejection | RBO↑ |
|---|---|---|---|---|
| text-davinci-003 | **0** | 280 | 0 | 72.30 |
| gpt-3.5-turbo | 14 | 153 | 7 | 81.49 |
| gpt-4 | **0** | **1** | 11 | **82.08** |

Table 10: **Analysis of model stability on TREC.** *Repetition* refers to the number of times the model generates duplicate passage identifiers. *Missing* refers to the number of missing passage identifiers in model output. *Rejection* refers to the number of times the model rejects to perform the ranking. *RBO*, i.e., rank biased overlap, refers to the consistency of the model in ranking the same group of passages twice.

## F    Model Behavior Analysis

In the permutation generation method, the ranking of passages is determined by the list of model-output passage identifiers. However, we have observed that the models do not always produce the desired output, as evidenced by occasional duplicates or missing identifiers in the generated text. In Table 10, we present quantitative results of unexpected model behavior observed during experiments with the GPT models.

**Repetition.** The repetition metric measures the occurrence of duplicate passage identifiers generated by the model. The results indicate that ChatGPT produced 14 duplicate passage identifiers during re-ranking 97 queries on two TREC datasets, whereas text-davinci-003 and GPT-4 did not exhibit any duplicates.

**Missing.** We conducted a count of the number of times the model failed to include all passages in the re-ranked permutation output[9]. Our findings revealed that text-davinci-003 has the highest number of missing passages, totaling 280 instances. ChatGPT also misses a considerable number of passages, occurring 153 times. On the other hand, GPT-4 demonstrates greater stability, with only one missing passage in total. These results suggest that GPT-4 has higher reliability in generating permutations, which is critical for effective ranking.

**Rejection.** We have observed instances where the model refuses to re-rank passages, as evidenced by responses such as "*None of the provided passages is directly relevant to the query ...*". To quantify this behavior, we count the number of times this occurred and find that GPT-4 rejects ranking the most frequently, followed by ChatGPT, while the Davinci model never refused to rank. This finding suggests that chat LLMs tend to be more adaptable compared to completion LLMs, and may exhibit more subjective responses. Note that we do not explicitly prohibit the models from rejecting ranking in the instructions, as we find that it does not significantly impact the overall ranking performance.

**RBO.** The sliding windows strategy involves re-ranking the top-ranked passages from the previous window in the next window. The models are expected to produce consistent rankings in two windows for the same group of passages. To measure the consistency of the model's rankings, we use RBO (rank biased overlap[10]), which calculates the similarity between the two ranking results. The findings turn out that ChatGPT and GPT-4 are more consistent in ranking passages compared to the Davinci model. GPT-4 also slightly outperforms ChatGPT in terms of the RBO metric.

## G    Analysis on Hyperparameters of Sliding Window

To analyze the influence of parameters of the sliding window strategy, we adjust the window size and set the step size to half of the window size. The main motivation for this setup is to keep the expected

---

[9]In our implementation, we append the missing passages in their original order at the end of the re-ranked passages.
[10]https://github.com/changyaochen/rbo

| API | Instruction | Tokens | Requests | $USD |
|---|---|---|---|---|
| `text-curie-001` | Relevance generation | 52,970 | 100 | 0.106 |
| `text-curie-001` | Query generation | 10,954 | 100 | 0.022 |
| `text-davinci-003` | Query generation | 11,269 | 100 | 0.225 |
| `text-davinci-003` | Permutation generation | 17,370 | 10 | 0.347 |
| `gpt-3.5-turbo` | Permutation generation | 19,960 | 10 | 0.040 |
| `gpt-4` | Permutation generation | 19,890 | 10 | 0.596 |
| - rerank top-30 | Permutation generation | 3,271 | 1 | 0.098 |

Table 11: Average token cost, number API request, and $USD per query on TREC.

| Window size | Step size | nDCG@1 | nDCG@5 | nDCG@10 |
|---|---|---|---|---|
| 20 | 10 | 75.58 | 70.50 | 67.05 |
| 40 | 20 | 78.30 | 71.32 | 65.51 |
| 60 | 30 | 75.97 | 69.23 | 65.03 |
| 80 | 40 | 72.09 | 70.59 | 65.57 |

Table 12: Analysis on Hyperparameters of Sliding Window on TREC-DL19.

overhead of the method (number of tokens required for computation) low; i.e., most tokens in this setup are used for PG only twice. The experimental results are shown in Table 12[11]. The results show that the effect varies over a certain range of arrivals for different values of window size: window size=20 performs best in terms of nDCG@10, while window size=40 performs best in terms of nDCG@5 and nDCG@1. We speculate that a larger window size will increase the model's ranking horizon but will also present challenges in processing long contexts and large numbers of items.

## H  API Cost

In Table 11, we provide details on the average token cost, API request times, and USD cost per query. In terms of average token cost, the relevance generation method is the most expensive, as it requires 4 in-context demonstrations. On the other hand, the permutation generation method incurs higher token costs compared to the query generation method, as it involves the repeated processing of passages in sliding windows. Regarding the number of requests, the permutation generation method requires 10 requests for sliding windows, while other methods require 100 requests for re-ranking 100 passages. In terms of average USD cost, GPT-4 is the most expensive, with a cost of $0.596 per query. However, using GPT-4 for re-ranking the top-30 passages can result in significant cost savings, with a cost of $0.098 per query for GPT-4 usage, while still achieving good results. As a result, we only utilize GPT-4 for re-ranking the top 30 passages of ChatGPT on BEIR and Mr.TyDi. The total cost of our experiments with GPT-4 amounts to $556.

Since the experiments with ChatGPT and GPT-4 are conducted using the OpenAI API, the running time is contingent on the OpenAI service, e.g., API latency. Besides, the running time can also vary across different API versions and network environments. In our testing conditions, the average latency for API calls for gpt-3.5-turbo and gpt-4 was around 1.1 seconds and 3.2 seconds, respectively. Our proposed sliding window-based permutation generation approach requires 10 API calls per query to re-rank 100 passages. Consequently, the average running time per query is 11 seconds for gpt-3.5-turbo and 32 seconds for gpt-4.

## I  Results of Specialized Models

Table 13 lists the detailed results of specialized models on TREC and BEIR.

---

[11]Note that the results are obtained using `gpt-3.5-turbo-16k` API for managing long context.

| Method | DL19 | DL20 | Covid | NFCorpus | Touche | DBPedia | SciFact | Signal | News | Robust04 | BEIR (Avg) |
|---|---|---|---|---|---|---|---|---|---|---|---|
| BM25 | 50.58 | 47.96 | 59.47 | 30.75 | **44.22** | 31.80 | 67.89 | 33.05 | 39.52 | 40.70 | 43.42 |
| **Supervised** *train on MS MRACO* | | | | | | | | | | | |
| monoBERT (340M) | 70.50 | 67.28 | 70.01 | 36.88 | 31.75 | 41.87 | 71.36 | 31.44 | 44.62 | 49.35 | 47.16 |
| monoT5 (220M) | 71.48 | 66.99 | 78.34 | 37.38 | 30.82 | 42.42 | 73.40 | 31.67 | 46.83 | 51.72 | 49.07 |
| monoT5 (3B) | 71.83 | 68.89 | 80.71 | **38.97** | 32.41 | 44.45 | **76.57** | 32.55 | 48.49 | 56.71 | 51.36 |
| Cohere Rerank-v2 | 73.22 | 67.08 | 81.81 | 36.36 | 32.51 | 42.51 | 74.44 | 29.60 | 47.59 | 50.78 | 49.45 |
| **Unsupervised** *instructional permutation generation* | | | | | | | | | | | |
| ChatGPT | 65.80 | 62.91 | 76.67 | 35.62 | 36.18 | 44.47 | 70.43 | 32.12 | 48.85 | 50.62 | 49.37 |
| GPT-4 | **75.59** | **70.56** | **85.51** | 38.47 | 38.57 | **47.12** | 74.95 | **34.40** | **52.89** | **57.55** | **53.68** |
| **Specialized Models** *train on MARCO labels or ChatGPT predicted permutations* | | | | | | | | | | | |
| MARCO Pointwise BCE | 65.57 | 56.72 | 70.82 | 33.10 | 17.08 | 32.28 | 55.37 | 19.30 | 41.52 | 46.00 | 39.43 |
| MARCO Listwise CE | 65.99 | 57.97 | 66.31 | 32.61 | 20.15 | 30.79 | 37.57 | 18.09 | 38.11 | 39.93 | 35.45 |
| MARCO RankNet | 66.34 | 58.51 | 70.29 | 34.23 | 20.27 | 29.62 | 49.01 | 23.22 | 39.82 | 43.87 | 38.79 |
| MARCO LambdaLoss | 64.82 | 56.16 | 72.86 | 34.20 | 19.51 | 32.55 | 51.88 | 26.22 | 42.47 | 45.28 | 40.62 |
| ChatGPT Listwise CE | 65.39 | 58.80 | 76.29 | 35.73 | 38.19 | 40.24 | 64.49 | 31.37 | 47.61 | 48.00 | 47.74 |
| ChatGPT RankNet | 65.75 | 59.34 | 81.26 | 36.57 | 39.03 | 42.10 | 68.77 | 31.55 | 52.54 | 52.44 | 50.53 |
| ChatGPT LambdaLoss | 67.17 | 60.56 | 80.63 | 36.74 | 36.73 | 43.75 | 68.21 | 32.58 | 49.00 | 50.51 | 49.77 |
| deberta-v3-xsmall (70M) | 64.75 | 55.07 | 78.21 | 35.95 | 35.42 | 41.37 | 67.86 | 30.04 | 47.68 | 49.91 | 48.31 |
| deberta-v3-small (142M) | 67.85 | 58.84 | 78.88 | 36.55 | 36.16 | 40.99 | 66.66 | 30.29 | 49.17 | 49.73 | 48.55 |
| deberta-v3-base (184M) | 70.28 | 62.52 | 80.81 | 36.15 | 37.25 | 44.06 | 71.70 | 32.45 | 50.84 | 51.33 | 50.57 |
| **deberta-v3-large (435M)** | 70.66 | 67.15 | 84.64 | 38.48 | 39.27 | 47.36 | 74.18 | 32.53 | 51.19 | 56.55 | 53.03 |
| deberta-v3-large 5K | 70.93 | 64.32 | 84.43 | 38.66 | 40.72 | 46.28 | 73.88 | 31.93 | 52.24 | 55.89 | 53.00 |
| deberta-v3-large 3K | 70.79 | 63.91 | 84.21 | 38.73 | 39.83 | 45.74 | 74.41 | 31.92 | 52.29 | 57.42 | 53.07 |
| deberta-v3-large 1K | 69.90 | 64.81 | 83.38 | 38.94 | 36.65 | 44.46 | 71.96 | 30.19 | 50.73 | 53.74 | 51.26 |
| deberta-v3-large 500 | 69.71 | 62.00 | 83.54 | 37.23 | 33.68 | 44.56 | 70.48 | 28.70 | 45.64 | 42.67 | 48.31 |
| deberta-v3-large label 10K | 66.61 | 57.26 | 74.36 | 33.94 | 18.09 | 34.95 | 35.35 | 21.38 | 39.00 | 44.94 | 37.75 |
| deberta-v3-large label 5K | 68.98 | 61.38 | 80.73 | 35.68 | 20.48 | 37.34 | 54.63 | 24.25 | 36.94 | 51.13 | 42.64 |
| deberta-v3-large label 3K | 67.41 | 60.42 | 79.82 | 35.49 | 24.54 | 37.39 | 47.31 | 23.29 | 39.87 | 50.65 | 42.29 |
| deberta-v3-large label 1K | 65.55 | 60.93 | 77.70 | 33.29 | 23.36 | 36.38 | 31.10 | 21.71 | 34.28 | 38.31 | 37.01 |
| deberta-v3-large label 500 | 60.59 | 54.45 | 76.20 | 32.93 | 19.66 | 31.54 | 45.66 | 13.99 | 33.48 | 44.49 | 37.24 |
| deberta-v3-large monoT5-3B | 73.05 | 68.82 | 84.78 | 38.55 | 34.43 | 43.61 | 75.45 | 30.75 | 49.85 | 56.80 | 51.78 |
| deberta-v3-large chatgpt+label | 72.42 | 67.30 | 85.96 | 38.75 | 35.06 | 45.43 | 71.81 | 28.52 | 45.91 | 55.57 | 50.88 |
| deberta-v3-base label 10k | 65.66 | 59.84 | 71.63 | 34.65 | 16.53 | 32.59 | 34.65 | 22.64 | 37.60 | 44.02 | 36.79 |
| deberta-v3-small label 10k | 63.63 | 52.83 | 68.17 | 30.48 | 18.12 | 31.72 | 33.62 | 18.02 | 34.57 | 36.09 | 33.85 |
| deberta-v3-xsmall label 10k | 60.89 | 51.15 | 63.58 | 28.67 | 14.87 | 27.12 | 20.60 | 18.97 | 32.61 | 32.67 | 29.89 |
| **llama-7b** | 71.33 | 66.06 | 78.23 | 37.60 | 34.87 | 45.46 | 76.13 | 34.17 | 51.79 | 55.22 | 51.68 |
| vicuna-7b | 71.80 | 66.89 | 78.32 | 36.87 | 31.81 | 45.40 | 74.23 | 34.28 | 51.13 | 52.91 | 50.62 |
| llama-7b 10k label | 65.22 | 56.85 | 75.36 | 36.24 | 20.88 | 37.34 | 69.04 | 25.22 | 41.21 | 49.21 | 44.31 |
| llama-7b 5k label | 69.24 | 58.97 | 80.49 | 37.55 | 28.23 | 39.66 | 71.79 | 26.04 | 44.09 | 53.83 | 47.71 |

Table 13: **Results (nDCG@10) on TREC and BEIR.** Best performing specialized and overall system(s) are marked bold. The specialized models are fine-tined on sampled queries using relevance judgements from MARCO or ChatGPT.