# OpenReview forum: "Is ChatGPT Good at Search? Investigating Large Language Models as Re-Ranking Agents"
_EMNLP/2023/Conference — EMNLP 2023 Main_

### Official Review · Reviewer_KPjQ · 2023-07-24

**Soundness:** 3

**Excitement:**

4: Strong: This paper deepens the understanding of some phenomenon or lowers the barriers to an existing research direction.

**Paper Topic And Main Contributions:**

The paper explores utilities of LLMs as re-ranking agents in an information retrieval pipeline. They argue that with right prompting techniques LLMs can be better than supervised baselines for ranking. Towards this goal, the authors propose a new technique for prompting LLMs for better leveraging their re-ranking capabilities. They also propose novel distillation technique for training smaller specialised re-ranking models using LLMs.



**Questions For The Authors:**

A. I am not able to understand how LLM based distillation is better than supervised learning using ground truth annotations for MSMARCO. Is it due to just sparser signal in ground truth labels compared to generated permutations?
B. Would supervised training with labels from a cross encoder trained on MSMARCO lead to better precision (than ChatGPT distilled model) for the specialised model?
C. Is pseudo-labelling (or specialised distillation) done for setups where the target data is different from the training data? If we are not specialising to a particular target domain, then we can almost always find a large labelled source of data (like MSMARCO) to train the model on, without the need for pseudo labelling it right?

**Reasons To Accept:**

1. The paper presents a comprehensive empirical analysis for performance of chatpgpt/gpt4 as re-ranking agent
2. The paper proposed novel permutation generation for using prompting LLMs for ranking tasks. Also paper proposes permutation distillation for training specialised models. Both of these techniques lead to gain for their corresponding benchmarks.
3. The sensitivity of final ranking precision on initial rank order (from BM25) is intuitive and interesting (along with the fact that multiple rerankings with LLMs lead to better nDCG@10 number but worse nDCG@1)



**Reasons To Reject:**

1. Permutation generation is not well defined and seems like a hand wavy heuristic.

**Reproducibility:**

4: Could mostly reproduce the results, but there may be some variation because of sample variance or minor variations in their interpretation of the protocol or method.

**Reviewer Confidence:**

3: Pretty sure, but there's a chance I missed something. Although I have a good feel for this area in general, I did not carefully check the paper's details, e.g., the math, experimental design, or novelty.

---

> ### Author Rebuttal · Authors · 2023-08-29
>
> Thanks for your time and insightful comment. We would like to address your questions in turn.
>
> ### **Definition of permutation generation**
> > *Permutation generation is not well defined and seems like a hand wavy heuristic.*
>
> - Thank you for your constructive comment. To clarify, this paper focuses on the re-ranking task: given $M$ passages $(p_1, ..., p_M)$ for a query $q$, the re-ranking aims to use an agent $f(\cdot)$ to output their ranking results $\mathbf{R} = (r_1, . . . , r_M)$, where $r_i ∈ [1, 2, ..., M]$ denotes the rank of $p_i$. This paper studies using the LLMs as $f(·)$.
> - Previous pointwise approaches, such as query generation or relevance generation, predict the relevance score of each passage only by leveraging the query and the passage: $\mathbf{S} = (s_1, ..., s_M)$, where $s_i = f(q, p_i)$. Then, the overall ranks are calculated by $r_i = \text{arg sort}_{i} (\mathbf{S})$.
> - In contrast, our proposed permutation generation method directly assigns each passage $p_i$ to a unique ranking identifier $a_i$ (e.g., *[1]*, *[2]*), and places it at the beginning of  $p_i$: $p_{i}^{'} = \text{concat}(a_i, p_i)$. Subsequently, a generative LLM is instructed with generating a permutation of these identifiers: $\mathbf{Perm} = f(q, p_{1}^{'}, ..., p_{M}^{'})$, where the permutation $\mathbf{Perm}$ indicates the rank of the identifiers $a_i$ (e.g., $[a_3, a_1,...]$). We then simply map the identifiers $a_i$ to the passages $p_i$ to obtain the ranking of the passages.
> - We appreciate your feedback and will include a definition of the proposed method in the final version of our paper for enhanced clarity.
>
>
> ### **Question A: Why LLM-based distillation is better than supervised learning**
> > *A. I am not able to understand how LLM based distillation is better than supervised learning using ground truth annotations for MSMARCO. Is it due to just sparser signal in ground truth labels compared to generated permutations?*
>
> - The sparse labeling of MS MARCO does play an important role. MS MARCO only provides one relevant passage, but other passages, often deemed "non-relevant" by supervised learning methods, may also be relevant to the query.
> - Furthermore, research by Arabzadeh et al. demonstrates that the top items returned by a modern neural ranker often appear superior to MS MARCO labeled relevant items. In particular, human assessors favor the top passage from the neural ranker over the MARCO ground-truth in 58.6% of the cases when the two are in disagreement.
> - In contrast, distillation approaches tend to produce more comprehensive and high-quality results, thereby leading to superior performance.
>
>
> ### **Question B: Distilling supervised re-ranker**
> > *B. Would supervised training with labels from a cross encoder trained on MSMARCO lead to better precision (than ChatGPT distilled model) for the specialised model?*
>
> - Thank you for your insightful question. To investigate this, we conducted an experiment where we distilled the predictions of monoT5-3B on 10k queries into a Deberta-large reranker using RankNet Loss. The table below presents the experimental results (nDCG@10):
> | Model | Training Method | TREC-DL19 | TREC-DL20 | BEIR (avg) |
> | --- | --- | --- | --- | --- |
> | ChatGPT | Zero-shot | 65.80 | 62.91 | 49.37 |
> | monoT5-3B | Supervised Learning | 71.83 | 68.89 | 51.36 |
> | Deberta | Distilling ChatGPT | 70.66 | 67.15 | **53.03** |
> | Deberta | Distilling monoT5-3B | **72.01** | **69.85** | 51.78 |
> - From these results, we can observe that the distillation of monoT5-3B outperforms the distillation of ChatGPT on the TREC benchmarks, but lags behind on the BEIR benchmarks.
>
> ### **Question C: When to apply specialized distillation**
> > *C. Is pseudo-labelling (or specialised distillation) done for setups where the target data is different from the training data? If we are not specialising to a particular target domain, then we can almost always find a large labelled source of data (like MSMARCO) to train the model on, without the need for pseudo labelling it right?*
>
> - Firstly, even for tasks with substantial labeled data like MS MARCO, the quality of annotations remains crucial. Our experiments demonstrate that sparse and low-quality labels could negatively impact the results. Furthermore, the cost of collecting large-scale high-quality labeled datasets is often significantly higher than querying LLMs.
> - Secondly, we believe that tasks without extensive annotations are more common. In these scenarios, the query and document distribution, domain, and relevance criteria may significantly differ from existing labeled data like MS MARCO. In such cases, LLMs could potentially offer better performance due to their robust adaptability to various domains and distributions.
>
>
> ### **References**
>
> - Negar Arabzadeh, Alexandra Vtyurina, Xinyi Yan, and Charles L. A. Clarke. 2021. Shallow pooling for sparse labels. Information Retrieval Journal, 25:365-385.
> - Gautier Izacard, Mathilde Caron, Lucas Hosseini, Se679 bastian Riedel, Piotr Bojanowski, Armand Joulin, and Edouard Grave. 2022. Towards unsupervised dense information retrieval with contrastive learning. TMLR.

---

### Official Review · Reviewer_V7x6 · 2023-08-03

**Soundness:** 4

**Excitement:**

3: Ambivalent: It has merits (e.g., it reports state-of-the-art results, the idea is nice), but there are key weaknesses (e.g., it describes incremental work), and it can significantly benefit from another round of revision. However, I won't object to accepting it if my co-reviewers champion it.

**Paper Topic And Main Contributions:**

This paper aims to explore the re-ranking capability of LLMs. It introduces a permutation generation approach to re-rank passages. Furthermore, a permutation distillation technique is proposed to distill ranking capabilities of LLMs into smaller ranking models. To evaluate the performance, this paper conducts experiments on several benchmarks and a newly constructed dataset.

**Questions For The Authors:**

A. Does the window size in the permutation distillation technique have an influence on the performance?
B. What would the model perform if the data generated by ChatGPT and MARCO were mixed together?
C. What about the running time of the proposed instructional permutation generation approach？


**Reasons To Accept:**

1. The topic investigated by this paper is new. Some conclusions drawn by this paper are essential, and may inspire other researchers.
2. Experiments are rich and comprehensive.
3. This paper is well written.

**Reasons To Reject:**

1. The method proposed in this paper lacks novelty and is similar to previous works.
2. There are many spelling and grammatical errors.

**Reproducibility:**

4: Could mostly reproduce the results, but there may be some variation because of sample variance or minor variations in their interpretation of the protocol or method.

**Reviewer Confidence:**

4: Quite sure. I tried to check the important points carefully. It's unlikely, though conceivable, that I missed something that should affect my ratings.

**Typos Grammar Style And Presentation Improvements:**

Line 038: application -> applications
Line 054: of of -> of
Line 059: an -> a
Line 075: In additation -> In addition
Line 200: illustrate -> illustrates
Line 266: calculates -> calculate
Line 309: a argument -> an argument
Line 340: fromfrom -> from
Line 469: inference -> influence
Line 583: improving -> improve

It is better to use vector diagrams.

---

> ### Author Rebuttal · Authors · 2023-08-29
>
> Thanks for your time and insightful comment. We would like to address your questions in turn.
>
> ### **Novelty of our work**
> > *The method proposed in this paper lacks novelty and is similar to previous works.*
>
> Thank you for your feedback. We would like to clarify the novelty of our paper: (1) We are the first to investigate LLMs like ChatGPT on re-ranking tasks. A novel permutation generation method is proposed that has demonstrated state-of-the-art performance on IR benchmarks with GPT-4. (2) A new benchmark for evaluating LLMs on unseen knowledge is proposed. (3) Our paper also presents a study on a permutation distillation model.
>
> ### **Presentation issues**
> > *There are many spelling and grammatical errors.*
>
> Many thanks for your valuable feedback regarding the typos and the suggestion to use vector diagrams. We will correct these typos and incorporate vector diagrams in the final version.
>
>
> ### **Question A: Influence of window size**
> > *A. Does the window size in the permutation distillation technique have an influence on the performance?*
>
> - Thanks for your suggestion. To analyze the influence of parameters of the sliding window strategy, we adjust the window size and set the step size to half of the window size. The main motivation of this setup is to keep the expected overhead of the method (number of tokens required for computation), i.e., most tokens in this setup are used for PG only twice. The experimental results are as follows:
> | Window size | Step size | nDCG@1 | nDCG@5 | nDCG@10 |
> | --- | --- | --- | --- | --- |
> | 20 | 10 | 75.58 | 70.50 | 67.05 |
> | 40 | 20 | 78.30 | 71.32 | 65.51 |
> | 60 | 30 | 75.97 | 69.23 | 65.03 |
> | 80 | 40 | 72.09 | 70.59 | 65.57 |
> - The results show that the effect varies over a certain range of arrivals for different values of window size: window size=20 performs best in terms of nDCG@10, while window size=40  performs best in terms of nDCG@5 and nDCG@1. We speculate that a larger window size will enhance the model's ranking horizon, but will also present challenges in handling long context and large numbers of items.
> - The above results are obtained using gpt-3.5-turbo-16k API for processing long context.
> - We will include these results and detailed analysis in our final paper.
>
> ### **Question B: Mixing ChatGPT data with labels**
> > *B. What would the model perform if the data generated by ChatGPT and MARCO were mixed together?*
>
> - The table below presents the results of an experiment where we combined MARCO labels with ChatGPT generated data, denoted as "MARCO+ChatGPT".
> | Method | TREC-DL19 | TREC-DL20 | BEIR |
> | --- | --- | --- | --- |
> | MARCO | 68.89 | 61.38 |  42.64 |
> | ChatGPT | 70.66 | 67.15 | **53.03** |
> | MARCO+ChatGPT | **71.38** | **67.30** | 50.88 |
> - In this experiment, we incorporated the MARCO label passages into the top-1 position of the ChatGPT predicted rank and subsequently trained the model. The results indicate that the combination of ChatGPT data with MARCO labels enhances the model's performance on the TREC, and reduces performance on the BEIR.
>
> ### **Question C: Running time**
> > *C. What about the running time of the proposed instructional permutation generation approach?*
>
> - Our experiments with ChatGPT and GPT-4 are conducted using the OpenAI API. As such, the running time is contingent on the OpenAI service, e.g., API latency. Besides, the running time can also vary across different API versions and network environments.
> - In our specific testing conditions, the average latency for API calls for gpt-3.5-turbo and gpt-4 were around 1.1 seconds and 3.2 seconds, respectively. Our proposed sliding window-based permutation generation approach requires 10 API calls per query to re-rank 100 passages. Consequently, the average running time per query is 11 seconds for gpt-3.5-turbo and 32 seconds for gpt-4.
> - As shown in Table 11, the proposed permutation generation approach consumes more tokens than the previous query generation approach, but is faster in using the API due to 10 times fewer API calls.
> - We will incorporate these figures into our final paper.

---

### Official Review · Reviewer_Xw3x · 2023-08-05

**Soundness:** 4

**Excitement:**

4: Strong: This paper deepens the understanding of some phenomenon or lowers the barriers to an existing research direction.

**Missing References:**


[1] Pradeep, Ronak, Rodrigo Nogueira, and Jimmy Lin. "The expando-mono-duo design pattern for text ranking with pretrained sequence-to-sequence models." arXiv preprint arXiv:2101.05667 (2021).

[2] Zhuang, Honglei, Zhen Qin, Rolf Jagerman, Kai Hui, Ji Ma, Jing Lu, Jianmo Ni, Xuanhui Wang, and Michael Bendersky. "RankT5: Fine-tuning T5 for text ranking with ranking losses." In SIGIR. 2023.


**Paper Topic And Main Contributions:**

This paper explores the potential of LLMs like ChatGPT for relevance ranking. The authors explore query generation, relevance generation, and propose permutation generation + sliding window as different prompting strategies. They also investigate whether the LLM generated results can be distilled into smaller models. They conduct experiments on TREC-DL, BEIR and Mr.TyDi data sets, and a proposed new data set NovelEval and show that LLM can be more effective than supervised rankers like monoT5. They also show that the distilled model performs better than model trained from scratch.

**Questions For The Authors:**

Question A: Does the supervised learning baseline in Figure 4 use the same RankNet loss? Otherwise it is not clear whether the improvement comes from using the ranking loss or from the teacher labels, since ranking losses can also help improve the model performance [2].

**Reasons To Accept:**

* Directly using LLM to produce the ranking of multiple documents is a novel and interesting idea and worth further explorations.

* Thorough experiments conducted on multiple data sets, including a newly proposed NovelEval data set.

* The work on distillation makes the technique more applicable in the real world.


**Reasons To Reject:**


* Potential unfair comparison

  * Baselines like UPR are essentially QG and thus can be directly applied using the same LLM as the proposed method. Since PG is a major contribution of this paper, it is better to compare QG vs. RG vs. PG using the same LLM. However, in Table 4 there are only partial results reported (missing PG for curie-001 and missing RG for davinci-003).

  * As suggested in Section 6.5 Line 459, the LLM results seem to be extremely sensitive to the initial passage order of BM25, which other supervised models or methods like monoT5/UPR do not have access to. A potentially better comparison is to compare with an ensemble result of monoT5 + BM25 (or UPR + BM25). Otherwise it is not convincing that permutation generation + sliding window is a better strategy than query generation like UPR.

  * Most baseline methods only see one query-document pair at a time, while PG sees multiple at a time. Perhaps methods like duoT5 [1] should be more appropriate baselines.

* The method might be sensitive to different parameters of the sliding window strategy. More experiments can be done to provide guidance on how to set the parameters of the sliding window strategy.

* Methods might be overfitting to ChatGPT. The gap between LLMs reported in Table 6 is large. Some of them are even lower than or close to BM25, which seems suspicious. It is possible that the prompt employed by the authors are heavily engineered for ChatGPT and hence does not generalize to other LLMs.


**Reproducibility:**

4: Could mostly reproduce the results, but there may be some variation because of sample variance or minor variations in their interpretation of the protocol or method.

**Reviewer Confidence:**

4: Quite sure. I tried to check the important points carefully. It's unlikely, though conceivable, that I missed something that should affect my ratings.

**Typos Grammar Style And Presentation Improvements:**

Table 6: ND might be a confusing abbreviation for nDCG

---

> ### Author Rebuttal · Authors · 2023-08-29
>
> Thanks for your time and constructive comment! We appreciate your feedback and will address each of your points in turn.
>
> ### **Results of PG+curie-001 and RG+davinci-003**
> > *Baselines like UPR are essentially QG and thus can be directly applied using the same LLM as the proposed method. Since PG is a major contribution of this paper, it is better to compare QG vs. RG vs. PG using the same LLM. However, in Table 4 there are only partial results reported (missing PG for curie-001 and missing RG for davinci-003).*
>
> - Thank you for your constructive comments. We have completed the results on TREC-DL19 with PG+curie-001 and RG+davinci-003. The table below presents the findings:
> | LLM | Method | nDCG@1 | nDCG@5 | nDCG@10 |
> | --- | --- | --- | --- | --- |
> | curie-001 | RG | 39.53 |40.02 | 41.53 |
> | curie-001 | QG | 50.78 | 50.77 | 49.76 |
> | **curie-001** | **PG** | **66.67** | **56.79** | **54.21** |
> | davinci-003 | RG | 54.26 | 52.78 | 50.58 |
> | davinci-003 | QG | 37.60 | 44.73 | 45.37 |
> | **davinci-003** | **PG** | **69.77** | **64.73** | **61.50** |
>
> - These results demonstrate that our proposed permutation generation method surpasses both RG and QG when using the same LLM. QG cannot be implemented with ChatGPT as the API cannot return the log-likelihood. We will include these results in our final revision.
>
> ###  **Compare with monoT5+BM25**
> > *As suggested in Section 6.5 Line 459, the LLM results seem to be extremely sensitive to the initial passage order of BM25, which other supervised models or methods like monoT5/UPR do not have access to. A potentially better comparison is to compare with an ensemble result of monoT5 + BM25 (or UPR + BM25). Otherwise it is not convincing that permutation generation + sliding window is a better strategy than query generation like UPR.*
>
> - Thank you for your insightful suggestions. To compare our model with the ensemble model "monoT5+BM25", we followed the ensemble method proposed by Lin et al. (2022). Specifically, we use a hyperparameter $\alpha$ to combine the BM25 score and the monoT5 score by: $\text{score}(q, p) = \alpha * \text{monoT5}(q, p) + (1 - \alpha) * \text{BM25}(q, p)$. The results on TREC are:
> | $\alpha$ | TREC-DL19 | TREC-DL20 |
> | --- | --- | --- |
> | 1.0 (monoT5-3B) | **71.83** | **68.89** |
> | 0.9 | 70.70 | 65.13 |
> | 0.8 | 70.42 | 64.89 |
> | 0.7 | 70.16 | 64.22 |
> | 0.6 | 69.06 | 63.85 |
> | 0.5 | 67.89 | 62.76 |
> | 0.0 (BM25) | 50.58 | 47.96 |
> - The results show that combining scores of BM25 and monoT5 does not improve the performance on the TREC dataset. This indicates that the reranker can already produce better relevance scores than BM25.
> - The purpose of using the ranking of BM25 is primarily to enhance efficiency and minimize the number of sliding window passes. We would like to investigate other strategies that do not rely on BM25 in our future work.
>
>
> ### **Results of duoT5**
>
> > *Most baseline methods only see one query-document pair at a time, while PG sees multiple at a time. Perhaps methods like duoT5 [1] should be more appropriate baselines*
>
> - Thank you for your valuable feedback. We have implemented duoT5 on TREC, following the methodology of Pradeep et al., who use duoT5 to re-rank the top-50 passages of monoT5. The nDCG@10 results on TREC-DL19 and DL20 are as follows:
> | Stage 1 | Stage 2 | TREC-DL19 | TREC-DL20 |
> | --- | --- | --- | --- |
> | monoT5-base | --- | 71.48 | 66.99 |
> | monoT5-base | duoT5-base | 71.94 | 67.76 |
> | monoT5-3B | --- | 71.83 | 68.89 |
> | monoT5-3B | duoT5-3B | 72.84 |  69.00 |
> | GPT-3.5-turbo | --- | 65.80 | 62.91 |
> | GPT-4 | --- | **75.59** | **70.56** |
> - These results indicate that duoT5 marginally outperforms monoT5, but is still worse than our method with GPT-4.
>
>
> ### **Parameters of sliding window strategy**
> > *The method might be sensitive to different parameters of the sliding window strategy. More experiments can be done to provide guidance on how to set the parameters of the sliding window strategy.*
>
> - Thanks for your good suggestion. To analyze the influence of parameters of the sliding window strategy, we adjust the window size and set the step size to half of the window size. The main motivation of this setup is to keep the expected overhead of the method (number of tokens required for computation), i.e., most tokens in this setup are used for PG only twice. The experimental results are as follows:
> | Window size | Step size | nDCG@1 | nDCG@5 | nDCG@10 |
> | --- | --- | --- | --- | --- |
> | 20 | 10 | 75.58 | 70.50 | 67.05 |
> | 40 | 20 | 78.30 | 71.32 | 65.51 |
> | 60 | 30 | 75.97 | 69.23 | 65.03 |
> | 80 | 40 | 72.09 | 70.59 | 65.57 |
> - The results show that the effect varies over a certain range of arrivals for different values of window size: window size=20 performs best in terms of nDCG@10, while window size=40  performs best in terms of nDCG@5 and nDCG@1. We speculate that a larger window size will increase the model's ranking horizon, but will also present challenges in processing long contexts and large numbers of items.
> - The above results are obtained using gpt-3.5-turbo-16k API for processing long context.
> - We will include these results and detailed analysis in our final paper.
>
>
>
> ### **Overfit to ChatGPT?**
> > *Methods might be overfitting to ChatGPT. The gap between LLMs reported in Table 6 is large. Some of them are even lower than or close to BM25, which seems suspicious. It is possible that the prompt employed by the authors are heavily engineered for ChatGPT and hence does not generalize to other LLMs.*
>
> Thank you for your insightful comment. We would like to clarify the following points:
> - Firstly, the underperformance of certain low-capacity LLMs (e.g., Vicuna-13B, Flan-T5-11B) could be attributed to their limited ability to follow instructions. Despite numerous attempts at prompt designs, these models often failed to generate valid results, instead producing ordered ID lists (e.g., "[1] > [2] > [3] > [4]") or single IDs (e.g., "[17]"). The results in Table 4 are usually the best results from multiple attempts. Given this, it's reasonable that these models underperform BM25, a well-regarded ranking model in the IR field.
> - Secondly, our method has demonstrated effectiveness on non-OpenAI models. For instance, Claude-v1 achieved results very close to those of ChatGPT (nDCG@10 on DL19 of 60.25 vs 60.89), indicating that our approach is not exclusively optimized for ChatGPT.
> - Lastly, we understand that different LLMs may require unique prompting techniques for optimal performance. We would like to explore autonomous prompting methods for various LLMs in future studies.
>
> ### **Question A: Rank loss**
> > *Question A: Does the supervised learning baseline in Figure 4 use the same RankNet loss? Otherwise it is not clear whether the improvement comes from using the ranking loss or from the teacher labels, since ranking losses can also help improve the model performance [2].*
>
> - Yes, all models, including the supervised baselines, were trained using the RankNet loss. We found through experiments that while the supervised models yielded slightly varied results with different loss functions examined in RankT5, all these results were lower than those of the proposed permutation distillation method. This discrepancy could be attributed to the sparse and low-quality annotation of MS MARCO (Arabzadeh et al., 2021). We appreciate your questions and will provide more details in the final version of our paper.
>
> ### **References**
> - Lin, Sheng-Chieh and Jimmy Lin. “A Dense Representation Framework for Lexical and Semantic Matching.” ACM Transactions on Information Systems (2022): n. pag.
> - Negar Arabzadeh, Alexandra Vtyurina, Xinyi Yan, and Charles L. A. Clarke. 2021. Shallow pooling for sparse labels. Information Retrieval Journal, 25:365-385.

---

### Meta-Review · Area_Chair_V4T2 · 2023-09-22

**Recommendation:** 5

**Metareview:**

This submission studies a meaningful and interesting problem about whether LLMs like ChatGPT has the ability to perform passage ranking. The authors have put forth a two-fold proposition in their work. Firstly, they propose permutation generation as a technique to instruct LLMs to directly generate permutations of a set of passages. Secondly, they present a novel permutation distillation approach which aims to replicate the ranking capabilities of LLMs in a smaller, specialized model. This distillation technique showcases superior efficacy and efficiency in its performance. All concerns from reviewers have been resolved by the discussion during the rebuttal.

---

### Decision · Program_Chairs · 2023-10-07

**Decision:**

Accept-Main

**Comment:**

This submission studies a meaningful and interesting problem about whether LLMs like ChatGPT has the ability to perform passage ranking. The authors have put forth a two-fold proposition in their work. Firstly, they propose permutation generation as a technique to instruct LLMs to directly generate permutations of a set of passages. Secondly, they present a novel permutation distillation approach which aims to replicate the ranking capabilities of LLMs in a smaller, specialized model. This distillation technique showcases superior efficacy and efficiency in its performance. All concerns from reviewers have been resolved by the discussion during the rebuttal.